# On the Helminth Fauna of the Muskrat (*Ondatra zibethicus* (Linnaeus, 1766)) in the Barnim District of Brandenburg State/Germany

**DOI:** 10.3390/ani11082444

**Published:** 2021-08-20

**Authors:** Rolf K. Schuster, Peter Specht, Siegfried Rieger

**Affiliations:** 1Central Veterinary Research Institute, Dubai P.O. Box 597, United Arab Emirates; 2Institute for Parasitology and Tropical Veterinary Medicine, Free University of Berlin, Robert von Ostertag Str. 7, 14163 Berlin, Germany; 3Department of Forestry and Environment, University of Applied Sciences Eberswalde, Alfred-Möller-Str. 1, 16225 Eberswalde, Germany; p.specht@hnee.de (P.S.); srieger@hnee.de (S.R.)

**Keywords:** *Ondatra zibethicus*, helminths, Federal State of Brandenburg, Germany

## Abstract

**Simple Summary:**

The knowledge of the parasite fauna in a given territory is the basis for a successful control. Publications in Germany and abroad showed that muskrats are suitable intermediate hosts for *Echinococcus multilocularis*, a small tapeworm of red foxes and other carnivores with zoonotic potential. After the first detection of this tapeworm in the Brandenburg state of Germany, research started to investigate the distribution of this parasite in final hosts, but the question of identifying infections in intermediate hosts remained. Introduced more than 100 years ago from north America, muskrats were shown to be suitable intermediate hosts for this parasite. In own investigations, 130 muskrats were examined for internal parasites and eleven endoparasites were found. Examination showed that muskrats trapped in the Barnim district of Brandenburg are final hosts for seven intestinal trematodes and are intermediate hosts for four tapeworms of carnivores. The larval stage of *E. multilocularis* was not detected.

**Abstract:**

The muskrat is a neozoon species that has occupied many countries of continental North Europe after its introduction from north America as fur animals. Due to its burrowing activity it damages river and canal banks and structures of flood control. For this reason, the eradication of this alien species is recommended. Muskrats are also of parasitological interest since they can act as suitable intermediate hosts for *Echinococcus multilocularis.* On the other hand, little is known on the other helminths that infect muskrats. A total of 130 muskrats of different age groups trapped in different habitats in the Barnim district of the Brandenburg state by a professional hunter were examined for parasites and seven trematodes (*Echinostoma* sp., *Notocotylus noyeri, Plagiorchis elegans, Plagiorchis arvicolae, Psilosostoma simillimum, P. spiculigerum, Opisthorchis felineus* and four larval cestode species (*Hydatigera taeniaeformis, Taenia martis, Taenia polyacantha, Taenia crassiceps*) were detected. Larval stages of *E. multilocularis* were not found. *O. felineus* was found for the first time in muskrats in Germany. All the named parasites were present in Europe prior to the introduction of muskrats. With a prevalence of 48.9%, *Strobilocercus fasciolaris*, the larval stage of the cat tapeworm, *H. taeniaeformis*, was the most frequent parasite found in adult muskrats.

## 1. Introduction

The muskrat (*Ondatra zibethicus*) is a medium sized semiaquatic, herbivorous rodent which is native to North America and inhabits wetlands, river banks, irrigation channels, lakes, ponds, coastal areas and estuaries. To be protected from predators, muskrats place the entrance to their burrow below of the water level. In shallow water, muskrats construct lodges using plant material. Three female and two male muskrats had been introduced to Bohemia in Czech Republic as fur animal and for hunting purposes. The current muskrat population in Germany is based on these five animals (according to other sources five pairs of muskrats were released [1]). Due to its reproduction potential and the migration behavior of this animal species, muskrats had occupied territories in southern Germany already in 1928. Already two years later, muskrats were seen in territories that today belong to Thuringia, Saxony, Saxony-Anhalt and Brandenburg [2]. It is documented that muskrats were also released in England but were eradicated later on. Papers on helminth parasites of muskrats in Karelia [3] and in Siberia [4] suggest that muskrats were also introduced to Russia. Muskrats have also occupied territories in northern Mongolia, north-eastern China, North Korea and Honshu Island of Japan [1].

Due to its burrowing activity (it damages river and canal banks and structures of flood control) feeding damage in field crops and feeding on the protected freshwater pearl mussel and other mussel species, the muskrat is listed by the Inventory of Alien Invasive Species in Europe [5] as one of the worst invasive species in Europe and is recommended for eradication by the Bern Convention on the Preservation of European Wild Plants and Animals and their Natural Habitats [6].

In Germany, foxes, otters, minks, polecats, birds of prey and nowadays raccoons and raccoon dogs are the main predators and this predator-prey-relation reflects on the parasite fauna of muskrats.

Investigations on *Echinococcus multilocularis* in the Brandenburg state of Germany in the 1990s revealed that this zoonotic tapeworm can be found in red foxes [7] but the involvement of small mammals as intermediate hosts in the Brandenburg state was not investigated. Already Abuladze [8] mentioned findings of larval stages of. *E. multilocularis* in muskrats in the 1950th in Russia and listed 42 other intermediate host. Although muskrats are not the first choice of food for red foxes, they might be an indicator for the occurrence of the fox tapeworm since the lifespan of muskrats is longer than that of voles and mice the preferred food of foxes.

The aim of this paper was to examine the helminth fauna of muskrats in the Barnim district of the Brandenburg state of Germany under special attention to cestode metacestodes.

## 2. Materials and Methods

The Barnim district has the highest share of surface waters (5.8%) of all rural districts of the Brandenburg state. Large lakes (Werbellinsee, Grimnitzsee, Parsteinsee) and a number of smaller lakes and cut off meanders (Alte Oder) are relicts of the last ice age.

The northern part of the district has in addition, a number of melioration ditches and canals with slow running water as well as the flood plains of the river Oder. All these are suitable habitats for muskrats.

A total of 130 muskrats that were trapped with special body grip and baited vintage traps by a professional muskrat hunter in various habitats of the Barnim district were available for our research. Classification into age groups (adults, subadults, juveniles) was done by the hunter (Table 1).

The origin of muskrats was allocated to three major habitats (1. lakes, 2. cut-off meanders of the Alte Oder, including adjacent melioration canals and 3. running waters: river Oder and Finow canal) (Table 2).

Most of the carcasses were kept in a deep freezer until examination. In fully defrosted carcasses abdominal and thoracic cavities were opened by a scalpel and scissors and macroscopically examined for the presence of bladderworms. Special attention was paid to the liver as it is the preferred location of *E. multilocularis* larval stages. Cysts at the surface of the liver were removed and opened. The liver was then sliced into 0.5 cm slices to exclude parasites in deeper parenchyma levels. After removal of internal organs, stomach and caecum were separated, opened by scissors and its content was emptied in 1 L glass cylinders and mixed with 500 mL tap water. Mesenterium was removed from small intestines and the intestinal lumen was three times flushed with 60 mL of normal saline. The rinsing liquid was collected in a 1 L glass cylinder. Contents of stomach, small and large intestines were allowed to settle for 30 min and supernatant was discharged. Water was added and this procedure was repeated until the supernatant became transparent. The sediment was transferred into Petri dishes and checked under a stereoscopic microscope.

For the photographic depiction, temporary mounts were produced. For this, cestode larvae or their anterior ends (in case of strobilocerci) were put for 20 min into 40 °C artificial gastric juice on a magnate stirrer and after neutralization with phosphate buffer solution, few drops of bile were added to provoke the evagination of the scolex. The scolex was cut off right posterior to the suckers, was placed between two slides in apical position and was fixed and dehydrated in rising alcohol concentrations. Glycerin was used to clear the preparation. Staining with lacto-carmine was also done but did not reveal additional morphological structures. Trematodes were stained in lacto-carmine, dehydrated in rising alcohol concentrations and cleared in clove oil. Since echinostomatid trematodes tend to lose spines, for the depiction of the collar, an unstained specimen obtained from a freshly trapped muskrat was used.

## 3. Results

Of the 130 examined carcasses, 73 contained parasites. Thirty-eight muskrats harbored one parasite species. Two and three different parasite species were found in 27 and 7 muskrats, respectively, and four parasite species were found in a single animal only. Examination of 57 carcasses did not reveal parasites. In 28 (=53.8%) juveniles, 14 (45.1%) subadults and 15 (26.3%) adults no endoparasites were detected.

The endoparasite fauna of muskrats in the Barnim district of the Federal State of Brandenburg consisted of seven trematodes: *Echinostoma* sp. (Figure 1 and Figure 2)*, Notocotylus noyeri* (Figure 3)*, Plagiorchis elegans* (Figure 4)*, Plagiorchis arvicolae* (Figure 5)*, Psilotrema simillimum* (Figure 6)*, P. spiculigerum, Opisthorchis felineus* (Figure 7) and four larval cestodes (*Hydatigera taeniaeformis* (Figure 8 and Figure 9)*, Taenia martis* (Figure 10 and Figure 11)*, Taenia polyacantha* (Figure 12 and Figure 13)*, Taenia crassiceps* (Figure 14) (Table 2). Larval stages of *E. multilocularis* were not detected. All trematodes except the bile duct parasite *O. felineus* inhabited the small intestine. Strobilocerci of *H. taeniaeformis* were located in the liver while other metacestodes were found between intestinal lopes in the abdominal cavity.

Of the 73 positive muskrats, 43 were infected with one, 22 with two, 7 with three and one with four helminth species. A total of 40 animals showed Taeniidae metacestodes. Of the 30 cases of liver strobilocercosis, 23 occurred in adult, four in subadult and three in juvenile hosts. In two adult muskrats *H. taeniaeformis* larvae were found in combination with *T. martis* larvae. Eight muskrats (five adults, two subadults and one juvenile) harbored *T. martis*. In addition to the above mentioned mixed infection with *H. taeniaeformis* strobilocerci, another animal showed a combination of *T. martis* and *T. polyacantha*. *T. polyacantha* and *T. crassiceps* metacestodes were found in four and one animal, respectively. All these five muscrats were classified as adults.

## 4. Discussion

The parasite spectrum of muskrats in Europe is relatively poor compared to that of North America. In publications on muskrat parasites from Canada and four US states 35 to 54 different endoparasite species were listed [9]. Examination of helminth fauna of muskrats in the state of Saxony-Anhalt in Germany (*n* = 80) showed the presence of 15 and 13 different species, respectively [10,11]. In the present material from the Brandenburg State eleven different helminth species were found.

Despite the presence of *E. multilocularis* in final hosts in the Brandenburg state of Germany [7,12,13,14], its larval stage was not detected in muskrats in our material.

The role of muskrats as intermediate hosts of *E. multilocularis* in Germany has been shown in Württemberg (southern Germany) [15] in an area that was known to be endemic for this parasite in foxes [16]. Larval stages of *E. multilocularis* were also found in muskrats hunted in Baden Wuerttemberg [17], Lower Saxony [18], in North Rhine Westphalia [19,20]. In addition, in France, Belgium and the Netherlands muskrats were found to be infected with *E. multilocularis* [21,22,23,24,25].

In our material, larval stages of *E. multilocularis* were not detected. Low prevalence of the adult cestode in final hosts and a relative low number of examined hosts might be the reason.

In the Brandenburg State *E. multilocularis* in red foxes is unevenly distributed. Prevalence in the high endemic focus reached 24% while in the surrounding low endemic area prevalence dropped to 5% [7]. Examination of raccoon dogs revealed a similar picture [13].

Taeniidae metacestodes were most often diagnosed in adult muskrats and there were also few cases of combination of two Taeniidae species. Thus, it is quite unlikely that muskrats ingest taeniid eggs accidently.

In accordance with other sources, *Strobilocercus fasciolaris*, the larval stage of the cat tapeworm, *Hydatigera taeniaeformis,* was the most frequent parasite in muskrats (Table 3). Apart from sources mentioned in Table 4, *S. fasciolaris* in muskrats in Germany was previously found by other authors [1,10,26,27,28]. Strobilocerci terminate with a small liquid filled bladder [29]. In Schleswig-Holstein, nearly all 670 examined muskrats were infected [30]. The pseudo segmented *S. fasciolaris* is located in up to cherry sized cysts in the liver parenchyma. Relaxed, it can reach a length of up to 459 mm. The scolex is armed with 30–34 rostellar hooks. Larger and smaller hooks measured 384–420 μm and 240–270, respectively.

Since muskrats spend most of the time in water, feed mainly on plants growing in water or on bank vegetation (bulrush, iris, sedges, reed grasses, water lilies and others) and only seldom leave the aquatic habitat, there is so far no explanation for the relative high metacestode infection rate. It cannot be excluded that muskrats cover their demand in minerals by ingesting carnivore feces. Other additional sources of calcium and phosphorus would be shells of snails, mussels or crabs. Cats may patrol the banks of water but avoid to enter the water and most probably do not prey on muskrats. Most of the infected muskrats in our material were trapped in lake habitats (Table 2). In Germany, *H. taeniaeformis* is the most frequent feline cestode. In a survey on parasites of feral cats in the Barnim district, between 6 and 30% harbored this tapeworm [31].

*Cysticercus talpae*, the larval stages of *T. mustelae* (syn. *T. tenuicollis*) is often situated in 3–5 mm oval shaped thin-walled cysts under the liver capsule of rodents and insectivores. Less often they can be found in the body cavity, under the skin or in the kidneys. The rostellum of the scolex is equipped with 44–50 or more hooks. Large and small rostellar hooks have nearly the same length, 170–190 and 210 μm, respectively. Apart from muskrats, *C. talpae* was found in bank voles, yellow necked mice, harvest mice, short tailed and common voles in Germany [32]. Final hosts of *T. mustelae* are weasels and other members of mustelids. This bladder worm was not found in own material.

Other metacestodes listed in Table 4 are situated in the body cavities not causing visible host reactions. *Cysticercus longicollis*, the larval stage of *T. crassiceps,* were found in the thorax of an adult muskrat trapped in one of the lakes. This larval stage multiplies in the intermediate host by budding. C. *longicollis* is a 2–4 mm long egg shaped, thin walled bubble with an invaginated scolex. Rostellar hooks in numbers between 30 and 36 were arranged in two circles and measured 180–197 and 130–151 μm, respectively. Rostellar hooks of both types had strikingly long blades. Budding toke place at the larger end, opposite of the scolex. The infected muskrat harbored 37 fully developed bladder worms, some of them were with buds. Previously, *C. longicollis* was detected also in a swelling between skeleton muscles [11]. Canids were listed as main and martens, badgers and cats as accidental final hosts for *T. crassiceps* [33]. In the Brandenburg state, *T. cassiceps* was found in raccoon dogs and red foxes in a prevalence of 2 and 5%, respectively [14].

*T. polyacantha* is another canid specific cestode with a large spectrum of rodents, including muskrats, as intermediate hosts. The larval stage of *T. polyacantha* was up to 10–12 mm long, with an invaginated scolex that bared up to 60 rostellar hooks arranged in two circles. The prevalence of *T. polyacantha* in final hosts, red fox and raccoon dog hunted in the neighboring Uckermark district added up to 40 and 13%, respectively [14].

*T. martis*, also known under its synonyms *T. intermedia, T. melesi, T. sibirica,* is a cestode of mustelids. Its larval stage with a white, bilaterally flatted, elongated body with frizzed margins was situated in the abdominal cavity without any viewable host reaction. The invaginated scolex was armed with 28 rostellar hooks arranged in two circles. Larger hooks were 175–195 μm and smaller hooks were 130–145 μm long, respectively. Both hooks stroke due to comparably small blades and relatively strong roots. Apart from muskrats, the larval stage of *T. martis* was found in Germany also in beavers [34]. In Germany, the adult cestode was previously found in stone martens [15], otters [35] and beech martens [36].

Allegedly, larvae of *T. pisiformis* and tetrathyridia of *Mesocestoides* sp. that were found in the liver parenchyma [10] most probably, were misdiagnosed early stages of *S. fasciolaris*, since *C. pisiformis* is usually found in the mesenterium and *Mesocestoides* tetrathyridia are located free in abdominal and thoracic cavities of intermediate hosts [15]. In our opinion, these stages could also be *C. talpae*.

As far as we know, the recent muskrat population in Germany go back to five to ten specimen that were released in Bohemia [1] and examination showed that the trematode, *Quinqueserialis quinqueserialis*, seems to be the only species that has traveled with infected muskrats from America to Europe and was able to establish the life cycle in the new habitats. *Qu. quinqueserialis* is a frequent parasite of muskrats in America and planorbid snails of the genus *Gyraulus* are the intermediate hosts. Experimental infections showed that at least 15 rodents are susceptible for an infection and produced fertile trematodes [37]. In Germany, this notocotylid species was found in muskrats in Saxony-Anhalt [10,11] and Lower Saxony [38]. In addition, apart from muskrats, *Qu. quinqueserialis* was also found in a brown rat in Saxony-Anhalt [39]. It has not been detected in our material from Barnim district.

All other diagnosed trematodes were already present in Europe prior to the introduction of muskrats and were obtained from other hosts.

Two species of the genus *Psilotrema, P. simillimum* and *P. spiculigerum* were the most frequent helminth found in muskrats of the Barnim district. Findings of *Psilotrema* spp. were more frequent in muskrats originating from lakes and cut off meanders of the Alte Oder (Table 2). Both species are primarily intestinal trematodes of water fowl and were originally described from white eyed pochards [40]. Smew, Pallas’s gull and non-fish eating anseriform birds were later listed as final hosts [41,42]. Both *Psilotrema* species were also found in European water voles and muskrats in the European part of the former Soviet Union [43]. *P. simillimum* differs morphologically from *P. spiculigerum* by strikingly larger pharynx and ventral sucker compared to the oral sucker. Species differentiation in deep frozen carcasses were difficult and for this reason both species were combined as *Psilotrema* spp. in Table 2. Prosobranch snails of the genus *Bithynia* act as intermediate hosts for both fluke species [41], cercariae encyst on water plants at the surface of the water and are ingested accidentally by the final hosts. It has to be underlined that *B. leachi* is a very rare snail species in Germany but was frequently seen in the Finow canal of Barnim district, while *B. tentaculata* is a ubiquitous species [44]. In addition, other sources [10,45,46] reported findings of *P. pharyngeatum* and *P. marki* in muskrats. Both names are junior synonyms for *P. simillimum* and *P. spiculigerum*, respectively [47].

*Plagiorchis elegans* is a trematode with a wide spectrum of final hosts. *P. elegans* was found in six lizard [48] and 25 bird species belonging to different orders as final hosts [42]. It also occurs in 18 rodent species including the muskrat [41]. More than ten *Plagiorchis* species were named as muskrat parasites. However, the validity of many of those species is doubtful because experimental studies showed that this species showed high morphological variability depending on final hosts [49,50,51,52]. Freshwater snails, *Lymnea stagnalis* and *Stagnicola palustris* are first and a variety of aquatic insect larvae and crustaceans act as second intermediate hosts [53,54]. Final hosts become infected when the accidently ingest infected larval stages of lake flies and other small water arthropods. *P. elegans* was more often diagnosed in muskrats from lakes and cut off meanders (Table 2).

The other species of the genus *Plagiorchis*, *P. arvicolae*, differs strikingly form *P. elegans* by its ellipsoid shape and massive vitellaria that cover most of the body. Most of the publications on this parasite species originate from former USSR where it was found in 11 rodent species in Russia, Belarus, Ukraine, Georgia, Azerbaijan, Armenia and Kazakhstan [55]. In addition, muskrats were mentioned as final hosts although *P. arvicolae* is a typical parasite of the northern water vole. *P. arvicolae* has lymnaeid snails as first and aquatic insect larvae (caddisfly, chironomids, dragonflies, damselflies) as second intermediate hosts. Metacercariae were also detected in their sporocysts [43,55].

*Notocotylus noyeri* is another trematode that was more often found in the northern water vole and is less frequent in other vole species [43]. It is one of the three *Notocotylus* species in Europe that occurs in small rodents. All other European species of this genus *Notocotylus* have birds as final hosts [56,57]. The life cycle of *N. noyeri* was studied in Germany and planorbid snails (*Bathyomphalus contortus*, *Anisus leucostomus* and *A. vortex* were identified as intermediate hosts. Metacercariae encyst on solid items, under natural conditions on water snail shells [58]. *N. noyeri* can be considered as accidental parasite of muskrats since previously it has not been found in muskrats.

Contrary to *Psilotrema* spp. and *P. elegans* where most of the cases were detected in muskrats trapped in lakes and cut off meanders, seven of the 12 *Echinostoma* cases were found in animals that originated from running waters. Until now, the species inventory of the genus *Echinostoma* is disputed [59]. *Echinostoma* flukes armed with 37 spines on their head collar (*E. revolutum* group) contain 56 nominal species which are morphologically similar and difficult to distinguish. Müller [10] named *E. armigerum* as the species infecting muskrats in Germany. However, *E. armigerum* and also *E. coalitum* and *E. callawayensis* that were described as muskrat parasites in America are synonyms to *E. trivolis*, a species that occurs only in North America [60]. More recent molecular investigations confirmed that *E. trivolis* is the only *Echinostoma* species in muskrats in America [61]. In a previous study on muskrat helminths in the German State of Saxony-Anhalt [11], the detected *Echinostoma* species was named *E. echinatum* based on research work conducted in Bulgaria [62]. However, the validity of *E. echinatum* is disputed [63] and also in recent times not dissolved [59]. Various freshwater snails act as first intermediate hosts for *Echinostoma* spp. and final hosts become infected when ingesting metacercariae located in mussels, snails and amphibians.

*Opisthorchis felineus* from bile ducts of an adult female muskrat trapped in a cut off meander of the Oder River is a remarkable finding because *O. felineus* is a parasite of fish eating mammals including humans. It was the first record of this parasite in muskrats in Germany. The muskrat as host for *O. felineus* is mentioned in the Russian literature [43] as a non-specific parasite for rodents. Only one original source reported the finding of *O. felineus* in one out of 72 examined muskrats in Siberia [64].

So far, *O. felineus* was detected only in the north-eastern part of Germany and this fact is related to the restricted occurrence of its intermediate host, *B. leachi*. In the Barnim district, *O. felineus* was found in 26% of feral cats [30] and in 7.6% of red foxes [65]. The prevalence of *O. felineus* in foxes in western Brandenburg and Berlin was 8.9% [66] and 17.7% [67], respectively. Apart from cats and red foxes, this zoonotic parasite was found in the Brandenburg state also in dogs [68], raccoon dogs [14] and otters [35]. Final hosts get infected by ingesting raw cyprinid fish where metacercariae are located in fins, gills and flesh. In examination of 802 cyprinids caught in waters of Berlin and Brandenburg, the highest prevalence (>80%) of opisthorchiid metacercariae was found in id, bleak and silver bream [69].

No nematodes were detected in our material. The nematode spectrum of muskrats in Eurasia is poor and consists only of 10 species that are specific for terrestrial rodents [70]. With the exception of *Trichinella spiralis*, infection happens when eggs (*Hepaticola hepatica, Trichuris muris, Syphacia obvelata*) or larvae (*Heligmosomoides laevis, Heligmosomum costellatum, longistriata* spp., *Trichostrongylus retortaeformis*) are ingested. Infective larvae of *Strongyloides ratti* enter the host through the skin and *T. spiralis* infections are a result of feeding on carrion.

Previous examination of the helminth fauna of muskrats in Germany revealed *Trichuris* sp. [10,11,38], *Trichostrongylus retortaeformis* and *Heligmosomum polygurum* [10] and *Ascaris* sp. [10,11] in low prevalence.

## 5. Conclusions

Muskrats are herbivorous animals in the first place. While infection with *Psilotrema* spp. and *P. elegans* happens when muskrats accidently swallow infected small water insect larvae or crustaceans when feeding on water plants, infections with *Echinostoma* sp., *N. noyeri*, *P. arvicolae* and *O. felineus* happens when they feed on mollusks, amphibians or on dead fish as emergency food.

All the intestinal trematodes are related to water and are primarily parasites of other hosts. It has to be mentioned that except Taeniidae metacestodes, no other helminths (anoplocephalids, ascarids, trichurids and trichostrongylids) with a terrestrial life cycle were found. This indicates that the muskrats in the Barnim district mainly search for food in the water.

However, muskrats have to leave the aquatic environment to get infected with metacestodes of carnivore Taeniidae. The larval stage of *H. taeniaeformis* was the most frequent bladder worm not only in own material but also in other publications. The reason behind might be insufficient mineral content in water and marsh plants that forces muskrats to eat carnivore feces that contain a higher amount of calcium and phosphorous.

## Figures and Tables

**Figure 1 animals-11-02444-f001:**
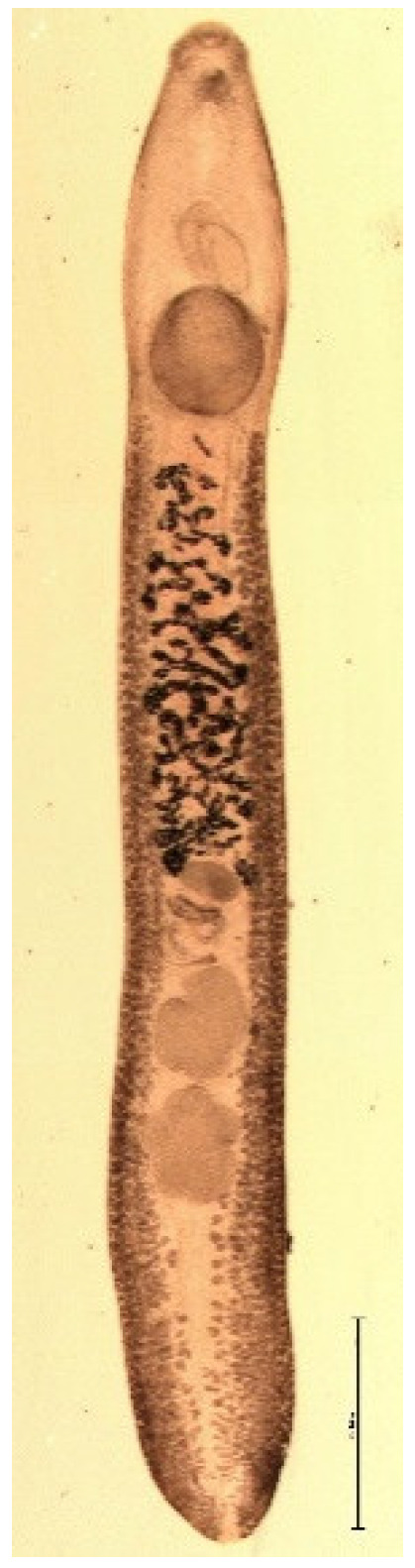
*Echinostoma* sp. total. Bar: 2 mm.

**Figure 2 animals-11-02444-f002:**
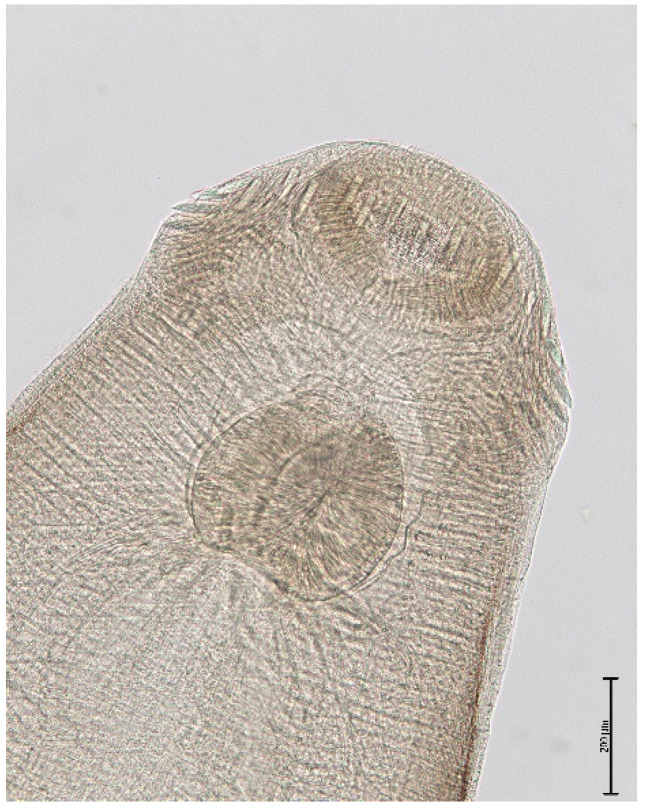
*Echinostoma* sp., anterior end with armed head collar, oral sucker and pharyx. Bar: 200 μm.

**Figure 3 animals-11-02444-f003:**
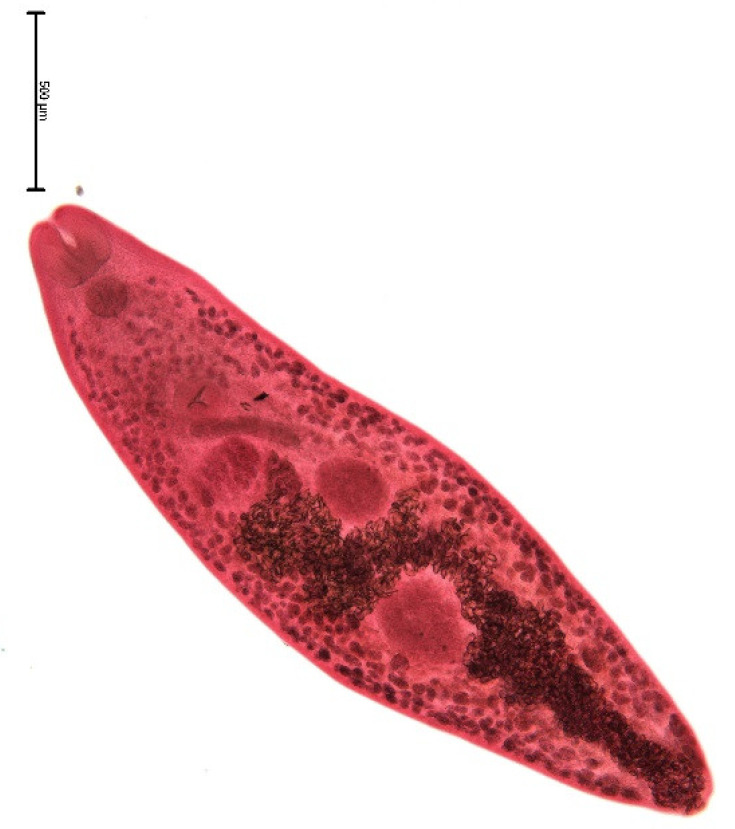
*Plagiorchis elegans.* Yolk glands form a bridge prior to the ventral sucker and reach the level of the pharynx. Bar: 500 μm.

**Figure 4 animals-11-02444-f004:**
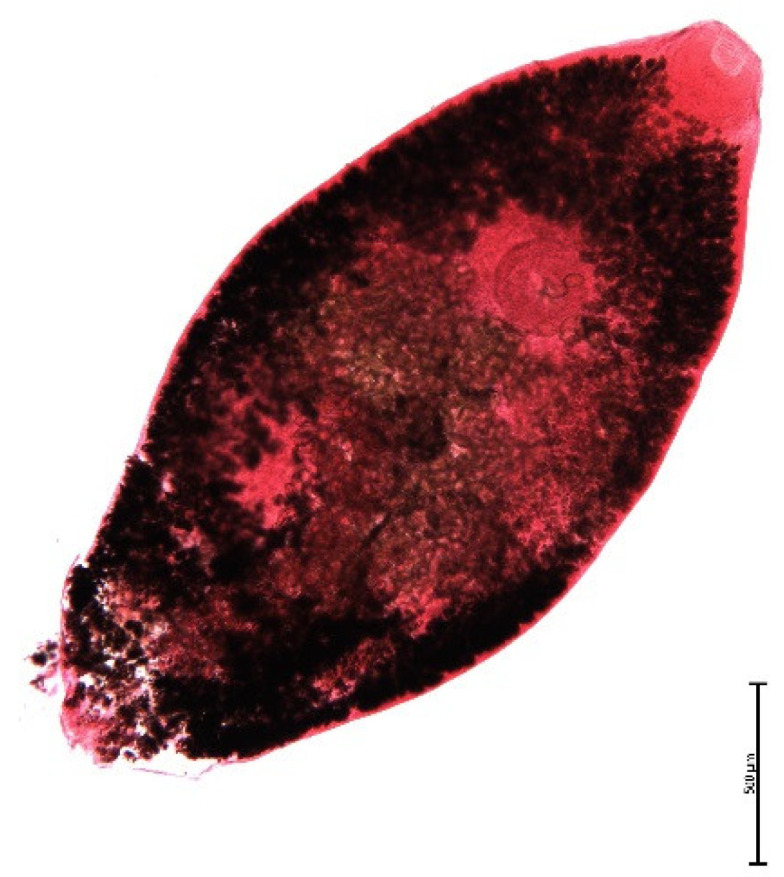
*Plagiorchis arvicolae.* Yolk glands fill most of the body. In unstained fresh specimens the trematode reminds an apple seed. Bar: 500 μm.

**Figure 5 animals-11-02444-f005:**
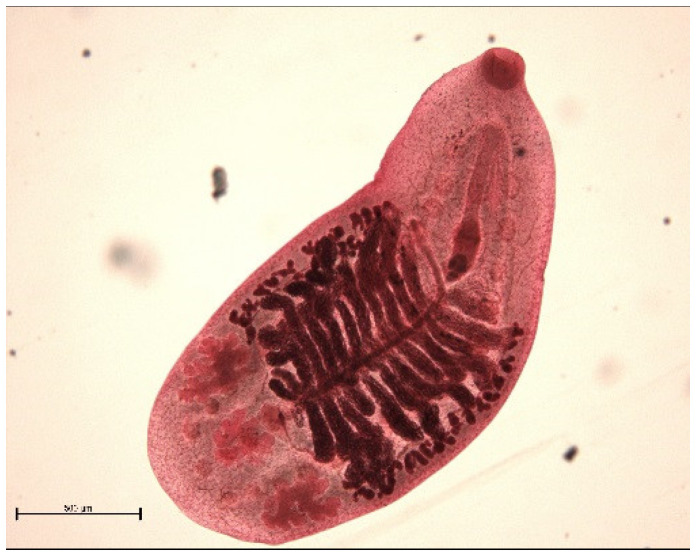
*Notocotylus noyeri.* There is no ventral sucker but there are three rows of ventral glands. Testes and ovary are situated at the posterior end. Branches of the uterus reach laterally to yolk glands. Bar: 500 μm.

**Figure 6 animals-11-02444-f006:**
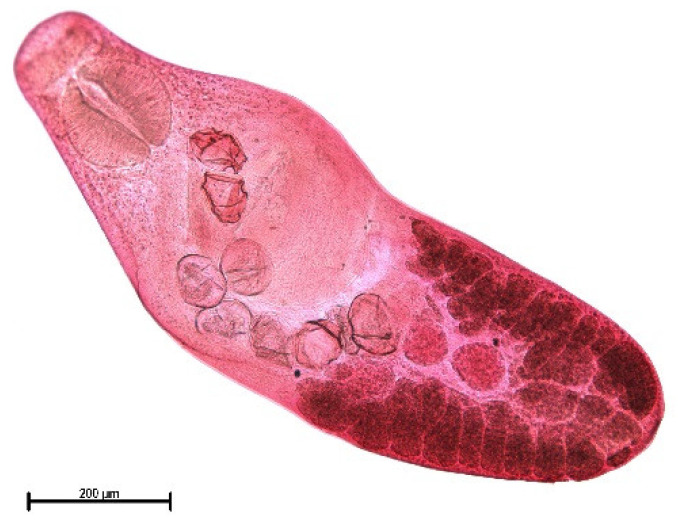
*Psilosostoma simillimum.* Pharynx and ventral sucker are strikingly larger than oral sucker. Bar: 200 μm.

**Figure 7 animals-11-02444-f007:**
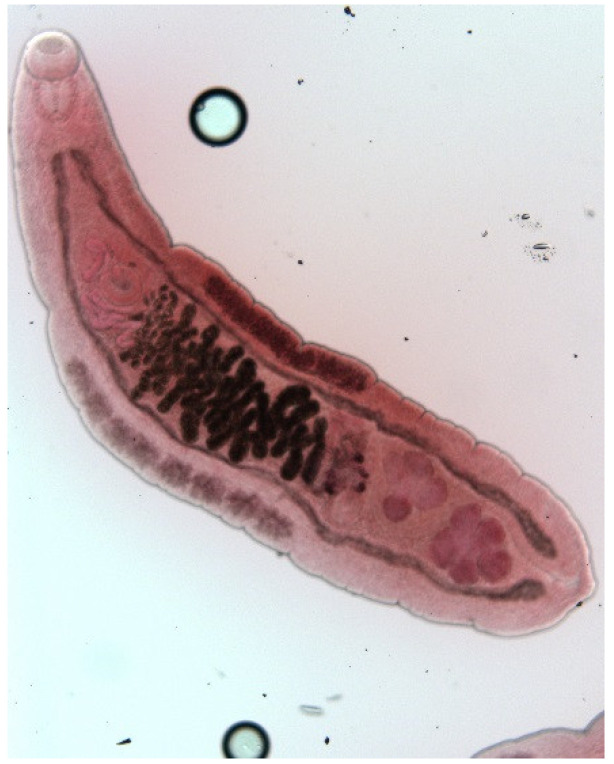
*Opisthorchis felineus.* Deeply lobed testes are situated at the posterior end. The uterus between ovary and ventral sucker does not overlap intestinal branches.

**Figure 8 animals-11-02444-f008:**
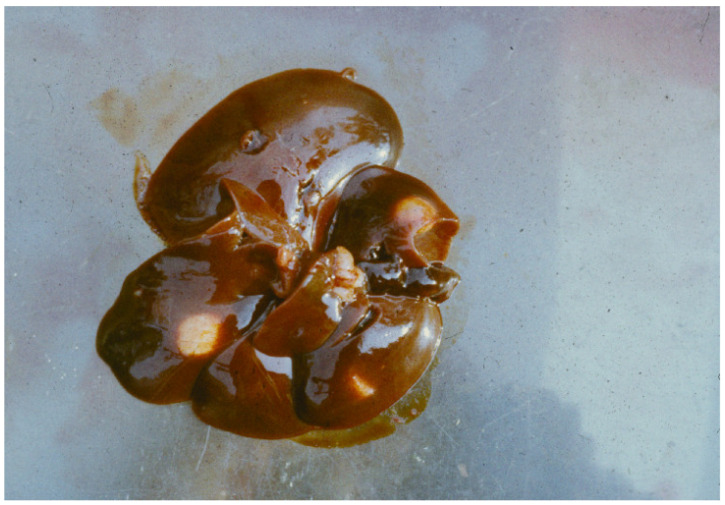
Three cysts of *Strobilocercus fasciolaris* the larval stage of *Hydatigera taeniaeformis* in the liver of a muskrat.

**Figure 9 animals-11-02444-f009:**
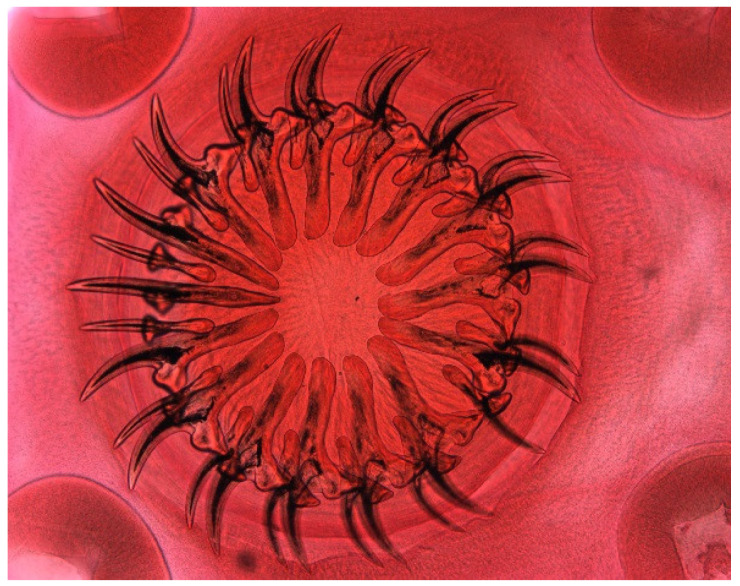
*H. taeniaeformis* scolex.

**Figure 10 animals-11-02444-f010:**
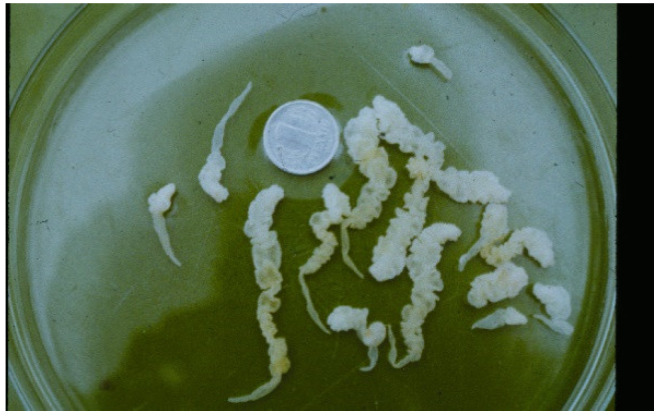
Larval stages of *Taenia martis* removed from abdominal cavity of a muskrat.

**Figure 11 animals-11-02444-f011:**
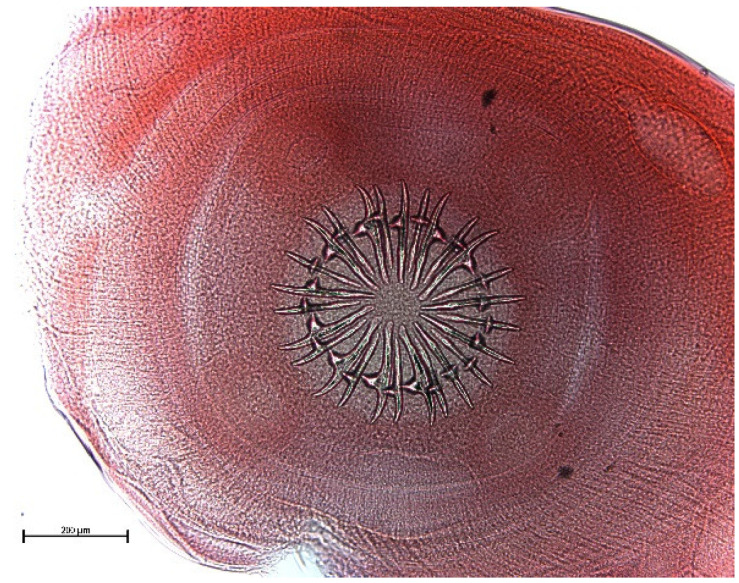
*T. martis* scolex. Bar: 200 μm.

**Figure 12 animals-11-02444-f012:**
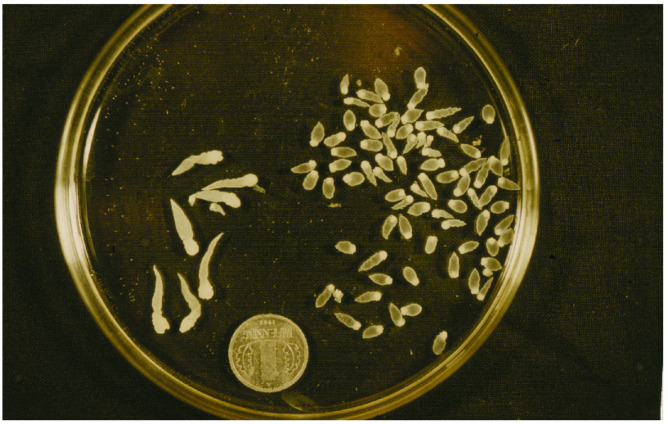
Larval stages of *Taenia polyacantha* removed from thorax of a muskrat.

**Figure 13 animals-11-02444-f013:**
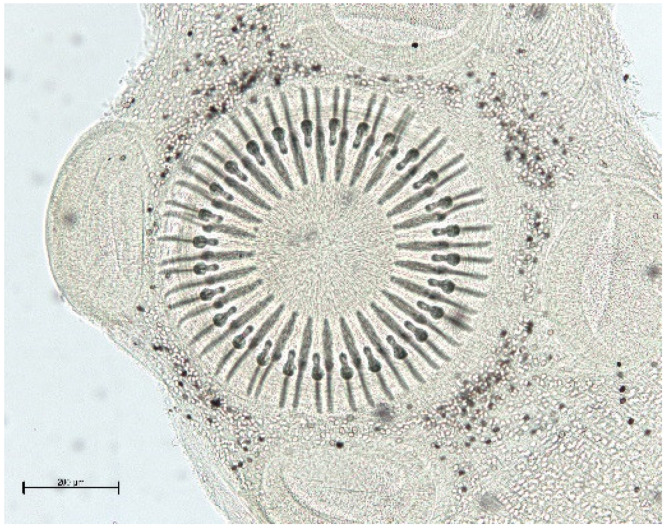
*T. polyacantha* scolex. Bar: 200 μm.

**Figure 14 animals-11-02444-f014:**
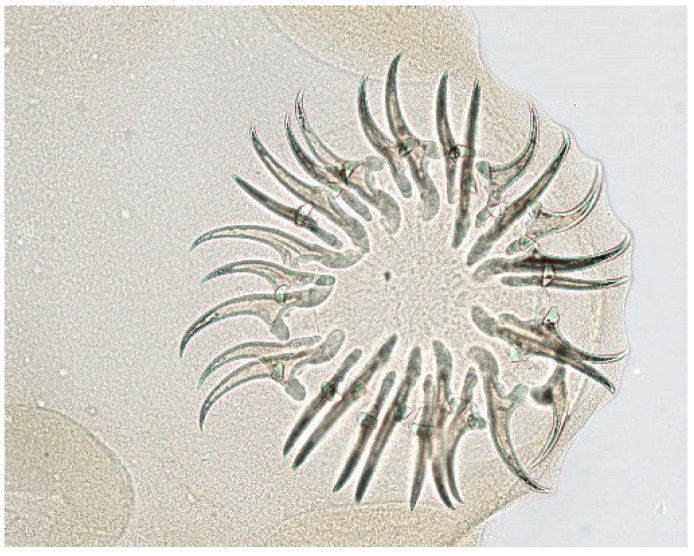
*Taenia crassiceps* scolex.

**Table 1 animals-11-02444-t001:** Age groups and habitats of muskrats.

Age Group	Habitats			Amount
	Cut off Meanders	Lakes	Running Waters	
Adults	8	24	15	47
Subadults	12	16	3	31
Juveniles	19	17	16	52
total	39	57	34	130

**Table 2 animals-11-02444-t002:** Helminth findings in muskrats originating from different habitats in the Barnim district.

Habitat	Number of Muskrats Infected with Parasites									
	*Echinostoma* sp.	*Plagiorchis* *elegans*	*Plagiorchis* *arvicola*	*Notocotylus* *noyeri*	*Psilotrema* spp.	*Opisthorchis* *felineus*	*Hydatigera* *taeniaeformis*	*Taenia martis*	*Taenia* *polyacantha*	*Taenia crassiceps*
Lakes (*n* = 57)	2	13	0	1	14	0	19	3	3	1
Cut off meanders (*n* = 39)	3	7	0	0	18	1	5	5	1	0
Running waters (*n* = 34)	7	1	1	0	1	0	6	0	0	0
Total (*n* = 130)	12	21	1	1	33	1	30	8	4	1

**Table 3 animals-11-02444-t003:** Prevalence of eleven parasite species found in muskrats in the Barnim district of Brandenburg state (range of intensity is given in brackets). The two species of *Psilotrema* are combined as *Psilotrema* spp.

Age Group	Parasites									
	*Echinostoma* sp.	*Plagiorchis* *elegans*	*Plagiorchis* *arvicola*	*Notocotylus* *noyeri*	*Psilotrema* spp.	*Opisthorchis* *felineus*	*Hydatigera* *taeniaeformis*	*Taenia martis*	*Taenia* *polyacantha*	*Taenia crassiceps*
Adults (*n* = 47)	8.5 (5–62)	17.0 (1–10)	2.1 (5)	0.0 (0)	17.0 (1–15)	2.1 (12)	48.9 (1–15)	10.6 (1–17)	6.4 (1–69)	2.1 (37)
Subadults (*n* = 31)	6.5 (1–5)	25.8 (1–8)	0.0 (0)	0.0 (0)	35.5 (1–35)	0.0 (0)	12.9 (1–3)	6.5 (1)	3.2 (5)	0.0 (0)
Juveniles (*n* = 52)	11.5 (1–20)	9.6 (1–2)	0.0 (0)	1.9 (3)	25.0 (1–18)	0.0 (0)	3.8 (1–2)	1.9 (1)	0.0 (0)	0.0 (0)
total (*n* = 130)	9.2 (1–62)	16.2 (1–10)	0.8 (5)	0.8 (3)	24.6 (1–35)	0.8 (12)	23.1 (1–15)	6.2 (1–17)	3.1 (1–10)	0.8 (37)

**Table 4 animals-11-02444-t004:** Prevalence of cestode larval stages in muskrats in Germany, Netherlands and Belgium.

Cestode Larval Stage	[10]	[15]	[30]	[11]	[18]	[25]	[24]	This Paper
	*n* = 630	*n* = 437	*n* = 670	*n* = 80	*n* = 991	*n* = 1726	*n* = 657	*n* = 130
	Germany	Germany	Germany	Germany	Germany	Netherlands	Belgium	Germany
*Echinococcus multilocularis*	0	1.8	0	0	4.1	0.1	22.1	0
*Hydatigera taeniaeformis*	33.17	48.1	99.6	63.75	42.3	44.8	65.8	23.1
*Taenia martis*	3.02	48.3	0	18.75	3.4	6.1	22.2	10.6
*Taenia crassiceps*	0.48	0.9	1.1	6.25	2.2	0.3	0.9	0
*Taenia polyacantha*	0.32	7.3	0	1.25	0.4	0.2	2.6	6.4
*Taenia pisiformis*	2.38	0	0	0	0	0	0	0
*Taenia mustellae*		4.8	0	0	0	0	0	0
*Mesocestoides* sp.	3.79	0	0	0	0	0	0	0

## Data Availability

Not applicable.

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
