# Peer review of "On the Helminth Fauna of the Muskrat (Ondatra zibethicus (Linnaeus, 1766)) in the Barnim District of Brandenburg State/Germany"

_animals, 2021, doi:10.3390/ani11082444_

Round 1

Reviewer 1 Report

This is an interesting manuscript with sound information and excellent photographic figures. This Reviewer (TR) has no doubt that the manuscript should be published. However, TR would wish the authors to consider a number of minor issues which might perhaps be improved.

The interpretation that muskrats are attracted to feline faeces because of the high prevalence of Hydatigera larval forms might be erroneous if filter-feeding molluscs enrich taeniid eggs from water. TR did not find any evidence in literature on this happening, but this possibility should be considered.

TR recommends not to italicize, neither to capitalize the  names of larval forms of tapeworms, such as strobilocercus and cysticercus in order to avoid confusing them with scientific species names.

Minor issues to consider:

Line 3: 1766)]

23: North

72: 1990s (?)

89: meadows by the river (?)

94: Line space after

Table 2: Please increase the size

96: three

105: 0.5 cm (??)

109: three

111: settle, how long?

112: Please, consider deleting "And".

127: taeniaeformis (Fig. (add space)

134: North

138: In the present material

142, 150, 176: Please consider "our" instead of "own".

160: Tab. 4 (?)

162: Delete "that".

Table 3: Plagiorchis elegans range [1-10]

180, 187: mustelae 

190: Tab. 4 (?)

211: mustelids

212: is (?)

227: specimens

231: muskrats

237: our

243 (Tab. 2)   primarily

250: was

252: [41], cercariae (add comma)

Tab. 4: et al. (add points: Baumeister and Borgsteede) Mathy et al. 2009

269: they

284: [56, 57]

288: an accidental 

Author Response

Regarding the high prevalence of larval stages of H. taeniaeformis there is no other explanation, since first of all cats do not tend to defaecate into water and in previous studies by RKS, there were no filter feeding molluscs in waters due to chemical pollution. Also there are observations that elks in northern Russia suffer from hydatidosis because they eat wolf faeces in late winter, to meet growing mineral demands for the developing foetus in females and for the antlers in males.  This was published by a Russian parasitologist with the name of Kozlov in the mid-1980th but unfortunately we don’t have this reference anymore.

With regards to the opinion of TR not to italicize Latin terms for cestode larval stages, the authors disagree, since Strobilocercus fasciolaris and others are synonyms for the adult stage. They should not italicised, when standing alone, like strobilocercus or cysticercus.

All the minor issues marked by the reviewer were changed:

89: river meadow was changed with flood plains

Tab. 2: is a technical matter how to present this table to make text bigger, maybe by putting the table in landscape format.

Reviewer 2 Report

This is an excelent faunistic work. I regret that no drawings were included, but this does not lower my generally high evaluation. 

Author Response

Yes, drawings could have been done for the trematodes but we preferred to present photographs

Reviewer 3 Report

Dear authors, 

Thank you for this interesting work. 

Here you are my comments for your consideration:

1- Title: good

2- Abstract and Keywords: good 

3- Introduction: good

4- Materials and methods:

  • please, add the ethical approval/statement.
  • please, add methods of parasitological examination ( e.g. preservation, 
  • , fixation and staining)  with suitable references .
  • 4- Results:
  • *please add the scale bar for each image as written as they are not clear .
  • *please, the orientation of table 3 should be checked. 
  • 5- Discussion and conclusion: good
  • That is all for now. 
  • Thanks again. 
  • Regards

Author Response

Indeed, the description of bars in images at the given magnification are difficult to read. For this reason, we have given the size of the bar again in the caption of the figure. Unfortunately, two images were photographed with a camera without magnification bar. We also added to materials and methods. These are well known methods going back to the beginning of last century and there is no need for references.

The authors believe that there is no need for an ethical statement since eradication of the alien species muskrat is recommended (Ref. 5 and 6) and authors obtained the carcasses from a professional muskrat hunter who used recommended muskrat traps.